# Design of Multifunctional and Efficient Water-Based Annulus Protection Fluid for HTHP Sour Gas Wells

Qilin Liu [1], Xue Han [2,*], Jian Cao [1], Lang Du [1], Ning Jia [1], Rong Zheng [1], Wen Chen [3] and Dezhi Zeng [2,*]

1    PetroChina Southwest Oil & Gas Field Company North Sichuan Gas Mine, Mianyang 621000, China
2    State Key Laboratory of Oil & Gas Reservoir Geology and Exploitation, Southwest Petroleum University, Chengdu 610500, China
3    Research Institute of Natural Gas Technology, Southwest Oil & Gasfield Company, China National Petroleum Corporation, Chengdu 610500, China
*    Correspondence: hx19950106@163.com (X.H.); zengdezhi1980@163.com (D.Z.)

**Abstract:** In order to solve the corrosion problem of production string in the process of acidizing for the purpose of production, a new water-based annular protective fluid suitable for HTHP acid gas, including $H_2S$-$CO_2$ wells, was developed. Firstly, an appropriate deoxidizer, bactericide, and corrosion inhibitor shall be selected according to the production string of acid gas. In addition, the synergism between additives is evaluated. Then, by designing the additive ratio, the optimal formulation of the water-based annular protective fluid is determined. Finally, a high-temperature autoclave was used to evaluate the protective performance of the water-based annular protective liquid. The results showed that it is recommended to use water-based annular protective liquids prepared with clear water that comes easily from nature (rivers, etc.), which consist of a corrosion inhibitor, CT2-19C (30,000 ppm), BN-45 bactericide (2 g/L), and anhydrous sodium sulfite (3 g/L). The density of the water-based annulus protection liquid is 1.02 g/cm$^3$, and the freezing point is $-2.01$ °C. The dissolved oxygen content of water-based annulus protection fluids prepared with clear water in formation water shall be controlled within 0.3 ppm. The corrosion inhibition rate of water-based annular protective fluid in the liquid phase is higher than 90%, and the corrosion rate of P110SS steel in the gas–liquid phase is lower than the oilfield corrosion control index (0.076 mm/y).

**Keywords:** $CO_2$-$H_2S$ environment; water-based annulus protection liquid; corrosion inhibitor; deoxidizer; bactericide

## 1. Introduction

China's natural gas production is concentrated in the Sichuan Basin and the Tarim Basin. These high sulfur gas fields will produce acid gases, including $H_2S$ and $CO_2$, in the process of natural gas production [1–3]. The high content of $H_2S$ and $CO_2$ will seriously corrode the production of string and promote the failure of the production tubing [4–6]. Dong et al. pointed out that in an acidic environment, the production tubing at the bottom of the deep well contains water, which can cause internal corrosion [7]. This is mainly because carbon steel only contains one layer of FeS in the $CO_2$-$H_2S$ environment. The corrosion resistance of Cr-containing steel production tubing to $CO_2$-$H_2S$ is higher than that of carbon steel, which is mainly due to the fact that Cr-containing steel is rich in Cr and S elements from the outside to the inside. Cr-containing compound $Cr(OH)_3$ and Sulfide of Fe FeS are competitively deposited on the surface of 3Cr steel. The deposition of FeS inhibits the precipitation of $Cr(OH)_3$, which preferentially accumulates near the surface to form an inner layer, while $Cr(OH)_3$ is far away from the surface to form an outer layer [8].

In addition, the packer rubber cartridge of the production tubing will be severely corroded in the $CO_2$-$H_2S$ environment [9,10]. The effect of corrosion on the mechanical properties of AFLAS and FKM rubber was not obvious, but it was very significant on

HNBR rubber. FKM rubber has good corrosion and aging resistance, followed by AFLAS rubber, whereas HNBR rubber has the worst corrosion and aging resistance in the $CO_2$-$H_2S$ corrosive environment [11]. After the packer leaks the acid, gas will enter the annulus and cause pressure in the annulus, which will cause serious corrosion to the casing [12–14]. In addition, the annular space between the closed production tubing and oil casing, $CO_2$-$H_2S$ gas accumulation, bacteria propagation, and other conditions are relatively serious, resulting in thinning and corrosion of the casing wall thickness [15–17]. Therefore, it is necessary to add environmental control protection fluid in the annular space between the closed production tubing and oil casing. Annular protection fluid generally contains oil-based annular protection fluid and water-based annular protection fluid [18–21]. Oil-based annular protective fluid is usually based on white oil and engine oil, and then a small amount of corrosion inhibitor is added [22,23]. Water-based annular protective fluid is usually based on well pad water, and then different agents are added, such as corrosion inhibitors, deoxidizers, and bactericides [24,25]. Zeng et al. formulated an oil-based annular air protection solution suitable for a $CO_2$ gas injection environment, which is composed of white oil and a massive imidazoline inhibitor, and the corrosion inhibition rate was more than 80% [15]. Although many new annular protective fluids have been developed, there is still a lack of efficient annular protective fluids suitable for sour gas production wells. Among them, the oil-based annular air protection liquid will pollute the environment, and a large dose of additives is required to achieve a better protection effect, so it is urgent to prepare a high-efficiency water-based annular air protection liquid with a good corrosion inhibition effect that is environmentally protected. The paper aims to solve the corrosion problem of the oil casing annulus in high-acid recovery wells. A high-efficiency water-based annular protective fluid that fits HTHP sour gas wells has been developed and evaluated. First, the additives including deoxidizer, bactericide, corrosion inhibitor, pour point depressant, and density regulator were selected, and were then evaluated on the performance of added protection fluids on this basis. This paper's conclusion offers a basic guarantee for the safe operation of high-acid production wells.

## 2. Experiments

### 2.1. Environmental Analysis of Corrosion of Sour Gas Well Production String

According to the data of an oilfield in northwest Sichuan, the acid gas produced in wells mainly includes 1.5 MPa $H_2S$ and 3 MPa $CO_2$. The packer is easily leaked in an HTHP environment, resulting in the acid gas and trace oxygen in the formation quickly entering the annulus. Therefore, it is seen that the corrosion environment is harsh because of the high annulus pressure of 40 MPa and the high temperature of 160 °C, and Figure 1 exhibits the produced wells of high-temperature and high-pressure (HTHP) acid gas wells. The tail pipe of wellbore liner is made of nickel-base alloy with good corrosion resistance, and carbon steel P110ss is used above the setting section of the packer. The annular space between the closed production tubing and oil casing is easy to breed bacteria, including saprophytic bacteria, iron bacteria and sulfate-reducing bacteria, which further aggravates the corrosion. To balance the differential pressure between the upper and lower packers and protect the production tubing and oil casing from further corrosion, water-based annular protective fluids need to be added. The water-based annular protective liquid is composed of corrosion inhibitors, deoxidizers, and bactericides.

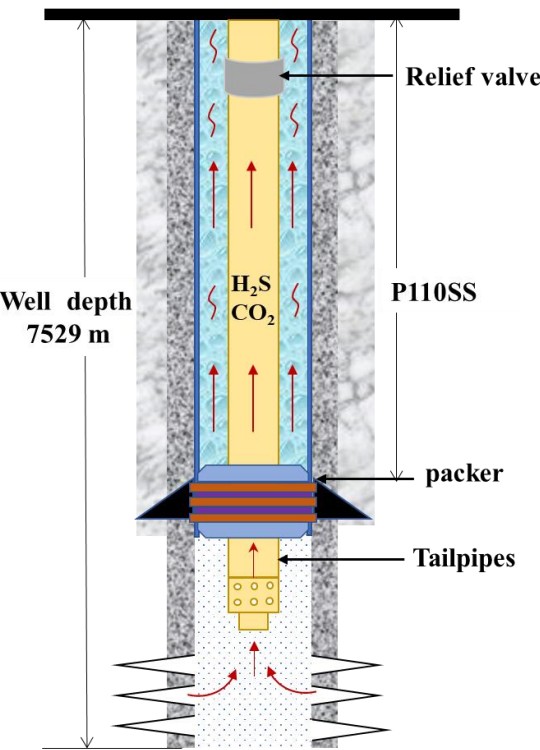

**Figure 1.** Schematic diagram of production well of HTHP acid gas well.

## 2.2. Experimental Consideration

Figure 2 exhibits the design thinking of water-based annular protective fluids fitted to HTHP sour gas wells. The design of water-based annular protective fluid mainly includes the selection and determination of additives, the selection and evaluation of additive synergy, the proportion design of additives, and the performance evaluation of water-based annular protective fluid. Among them, additives including corrosion inhibitors, oxygen scavengers, and fungicides are selected from mature products in the Chinese market.

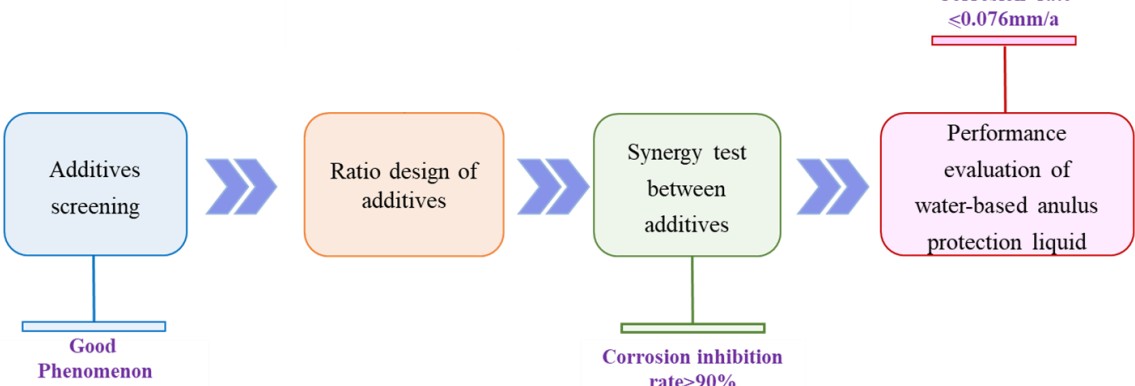

**Figure 2.** Design thinking for water-based annular air protective fluid.

## 2.3. Experimental Method

### 2.3.1. Physical and Chemical Properties Test

First, the water solubility of all active additives including, deoxidizer, bactericide, corrosion inhibitor, and formation water, needs to be considered. Therefore, the water solubility of active additives and formation water must be evaluated by SY/T5273-2014 (technical specifications and evaluating methods of corrosion-inhibitors for oilfield produced water). Then, the deoxidizers, bactericides, and corrosion inhibitors are screened

according to the standard SY/T5889-2010 (evaluation method for performance of deaerator), SY/T5757-2010 (general technical conditions for fungicides injected into oilfields), and SY/T205273-2014 [26,27]. Finally, on the basis of GB/T510-2018 (petroleum products' determination of freezing point), the freezing point of the annulus protective fluid is tested with a BSY-179D cryogenic flow tester (Dalian Beigang, Dalian, China ), and its density is tested with a mud hydrometer (NB-1).

### 2.3.2. Electrochemical Test

CorrTest CS305 electrochemical workstation (USA) is adopted. According to SY/T5273-2014 standard, three traditional electrode systems are used to conduct an electrochemical test of the corrosion inhibitor. Among them, P110ss steel (chemical composition seen in Table 1) is selected as the working electrode (WE) with a test surface area of 0.785 cm². The reference electrode is a double salt bridge saturated calomel electrode (SCE), and the platinum sheet as the counter electrode (CE). The saturated $H_2S$ aqueous solution (composition of formation water is shown in Table 2) was titrated into the saturated $CO_2$ aqueous solution as corrosive solution by titration method at a ratio of 1:2, and Thermo Scientific Dionex multifunctional ion chromatograph (Thermo Fisher Scientific, Waltham, MA, USA) was used to analyze water ion composition in formation water (Table 2). The electrochemical test was performed at a temperature of 60 °C. The electrochemical polarization curve test was compared with the open circuit potential $\pm$ 400 mV scan, and the scanning rate was 0.50 mV/s. The Tafel extrapolation method was used to fit the polarization curve measured by the experiment to determine the corrosion rate and corrosion inhibition rate.

**Table 1.** Chemical composition of P110SS steel (wt.%).

| C | Si | Mn | P | S | Cr | Mo | Ni | Nb | Ti | V | Fe |
|---|---|---|---|---|---|---|---|---|---|---|---|
| 0.27 | 0.26 | 0.60 | 0.009 | 0.003 | 0.50 | 0.60 | 0.25 | 0.05 | 0.02 | 0.005 | BaL |

**Table 2.** Chemical composition of simulated formation water (mg/L).

| Compounds | $Ca^{2+}$ | $Mg^{2+}$ | $Cl^-$ | $SO_4^{2-}$ | $HCO_3-$ | $Na^+/K^+$ |
|---|---|---|---|---|---|---|
| Content (mg/L). | 292.72 | 138.8 | 37,443.03 | 32.04 | 1868.2 | 31,489.37 |

### 2.3.3. HTHP Test

The self-designed HTHP autoclave made of C276 alloy with a volume of 8 L is used for weight-loss, as shown in Figure 3. The corrosion gas includes 1.5 MPa $H_2S$ and 3 MPa $CO_2$, and the total pressure is 40 MPa.

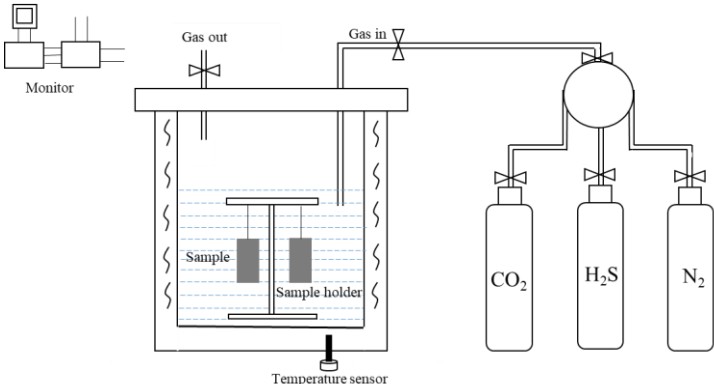

**Figure 3.** Schematic diagram of self-designed C276 alloy HTHP autoclave.

Four test pieces of P110ss steel are used for each group of experiments, and then the surface of the sample is polished using the silicon carbide sandpaper with 300 #,

600 #, 800 #, and 1200 # to remove surface scratches [22]. The surface of samples is cleaned with deionized water, absolute ethanol, and petroleum in turn, and dried out with cold air. The samples are then taken off and dried for 2 h, and an electronic balance with an accuracy of 0.1 mg was used for weighing. Before the experiment, $N_2$ gas is injected into the autoclave for 4 h to remove the oxygen. The simulated formation water is added into the HTHP autoclave, and $N_2$ gas is injected into autoclave for another 40 min for deoxygenation. HTHP autoclave is heated to 160 °C after sealing, 1.5 MPa $H_2S$ and 3 MPa $CO_2$ are introduced successively, and then $N_2$ gas is added to keep the total pressure at 40 MPa. What's more, the entire experimental duration was 168 h. After the experiment, the samples are taken out and dried with cold air. The samples used for weight-loss analysis need be immediately washed with deionized water, then dipped in the film removal solution (3% hydrochloric acid and 1% hexamethylene tetramine). The corrosion products on the surface of the sample need be wiped with a degreased cotton ball, then cleaned with distilled water and absolute ethanol in turn. In addition, the samples need to be dried by cold air to be weighed. The weighing results need be accurate to 0.1 mg, and the average corrosion rate $V$ shall be calculated using Equation (1) [28,29].

$$V = 87600\frac{\Delta m}{\rho S \Delta t} \tag{1}$$

where $V$ is the average corrosion rate (mm/y); $\Delta m$ is the weight loss of test samples before and after corrosion (g); $\rho$ is the density of steel (g/cm$^3$); $S$ is the surface area of test samples, (cm$^2$); $\Delta t$ is the corrosion time (h).

Finally, the surface morphology of corrosion product film was analyzed by FEI Quanta 450 scanning electron microscope (FEI, Hillsboro, OR, USA).

### 3. Results and Discussion

#### 3.1. Additive Research

3.1.1. Compatibility Test of Additives

Table 3 exhibits the compatibility results for the formation water and active additives. Figure 4 displays the compatibility of the formation water and active additives. It can be seen that only the YC-12 corrosion inhibitor dissolves in the formation water and produces a milky white flocculent. After being placed in a water bath at 60 degrees Celsius for 30 min, obvious stratification occurs, indicating that the YC-12 corrosion inhibitor has poor compatibility with formation water. Inorganic salts in the formation water reduce the solubility of long chain carboxylic acid in the carboxylic acid composite corrosion inhibitor, resulting in it having a lower solubility than the other two corrosion inhibitors, including Z-05 and CT2-19C. Therefore, the other two corrosion inhibitors have good compatibility with formation water.

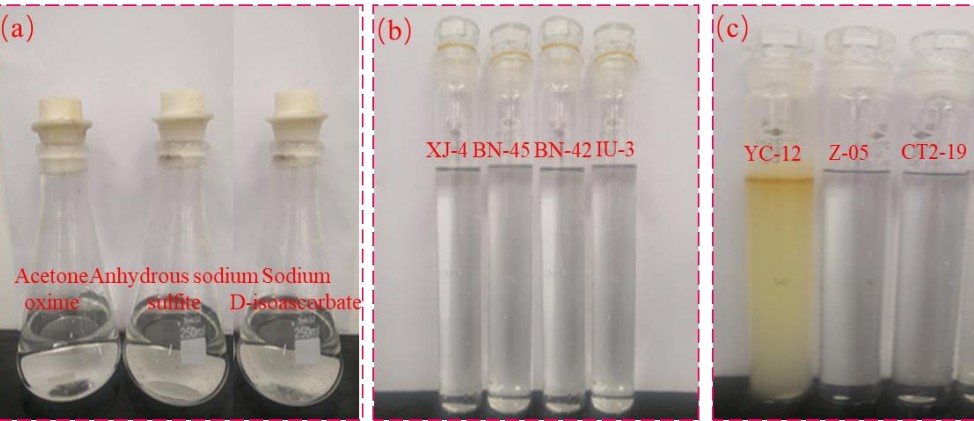

**Figure 4.** Compatibility of injected water with additives (**a**) deoxidizer; (**b**) bactericide; (**c**) corrosion inhibitor.

**Table 3.** Results of the compatibility of the injected water with additives.

| Additives | Type | | Phenomenon (30 min, 60 °C) | Evaluation Results |
|---|---|---|---|---|
| Deoxidizer | Acetone oxime | | homogeneous | good dispersion |
| | Anhydrous sodium sulfite | | homogeneous | good dispersion |
| | Sodium D-isoascorbate | | homogeneous | good dispersion |
| Bactericide | XJ-4 | | homogeneous | good dispersion |
| | BN-45 | | homogeneous | good dispersion |
| | BN-42 | | homogeneous | good dispersion |
| | IU-3 | | homogeneous | good dispersion |
| Corrosion inhibitor | YC-12 | carboxylic acid compound corrosion inhibitor | milky white flocs | good dispersion |
| | Z-05 | Imidazolines | homogeneous | good dispersion |
| | CT2-19C | Quaternary ammonium salts | homogeneous | good dispersion |

### 3.1.2. Deoxidizer Selection Test

Figure 5 displays the dissolved oxygen amounts of different deoxidizers in the formation water and the deoxidization rates of different deoxidizers. It can be seen that anhydrous sodium sulfite and sodium D-isoascorbate have good deoxygenation effects. Generally, organic fungicides such as acetone oxime can be charged onto the surface of the solution. The addition of salt water polymerizes the organic substances, reducing their solubility and affecting their deoxygenation performance. It is recommended to use anhydrous sodium sulfite and sodium D-isoascorbate for the next step of collaborative experimental evaluation.

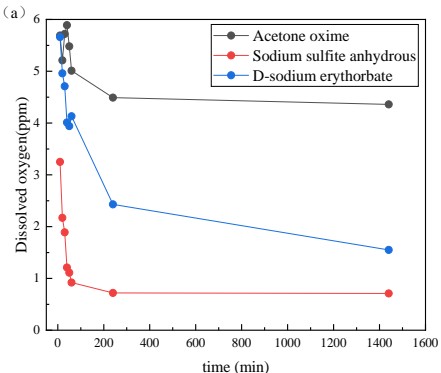 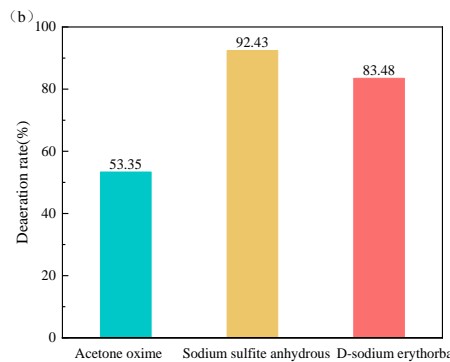

**Figure 5.** Dissolved oxygen of injected water containing different deoxidizers (**a**) and the deaeration rate of different deoxidizers (**b**).

### 3.1.3. Bactericide Selection

The bactericides are usually divided into oxidizing and non-oxidizing bactericides [30–32]. Due to the corrosion of oxidizing bactericide on steel, non-oxidizing bacteria XJ-4 modified quaternary ammonium salt bactericide, BN-45 glutaraldehyde composite bactericide, BN-42 glutaraldehyde bactericide, and IU-3 quaternary ammonium salt bactericide are selected for bactericide testing.

Figure 6 exhibits the bactericidal effect of different concentrations of bactericides on saprophytic bacteria, iron bacteria, and sulfate reducing bacteria. It can be seen that the bactericides BN-45 and BN-42 have a good bactericidal effect. Therefore, BN-45 and BN-42 bactericides are used with deoxidizers and corrosion inhibitors to assess synergy in the next step.

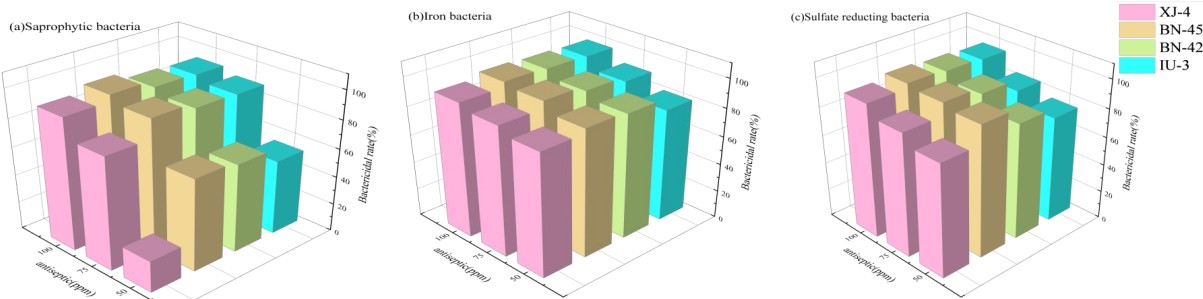

**Figure 6.** Sterilization rate of bactericides of different concentrations against bacteria. (**a**) saprophytic bacteria; (**b**) iron bacteria; (**c**) sulfate reducing bacteria.

### 3.1.4. Corrosion Inhibitor Selection

Figure 7 displays the self-corrosion current density and corrosion inhibition rate of P110ss steel in the formation water containing three corrosion inhibitors. Compared with other corrosion inhibitors, the self-corrosion current density of the CT2-19C corrosion inhibitor is the smallest, and its corrosion inhibition rate is 94.13%. Therefore, the CT2-19C corrosion inhibitor is used with deoxidizers to assess synergy in the next steps.

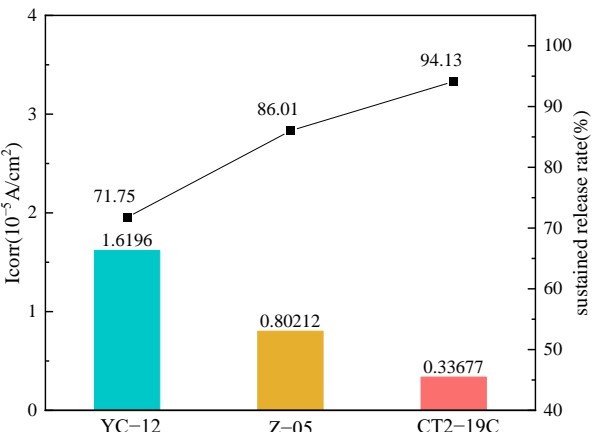

**Figure 7.** Self-corrosion current density and corrosion inhibition rate of P110ss steel in simulated corrosive water with three corrosion inhibitors YC−12, Z−05 and CT2−19C.

### 3.2. Protection Fluid Evaluation

### 3.2.1. Compatibility Experiment

The most important evaluation index of the annular space between the closed production tubing and the oil casing is the anti-corrosion performance of the production tubing and casing materials [33,34]. Therefore, the annular space between the closed production tubing and oil casing fluid should contain the corrosion inhibitor as the core, and the compatibility evaluation experiment of active additives such as deoxidizers, bactericides, and corrosion inhibitors should be carried out. Table 4 shows the compatibility test scheme and results of various additives. Figure 8 displays the compatibility results of each additive, and the muddying of the solution can be seen in groups one, five, and six. On this basis, the additive synergy evaluation experiment is carried out for groups seven and eight.

**Table 4.** Compatibility scheme and results of additives.

| Group | Corrosion Inhibitor | Deoxidizer | Bactericide | Phenomenon (30 min, 60 °C) | Evaluation Results |
|---|---|---|---|---|---|
| 1 | CT2-19C | Sodium D-isoascorbate | | muddy | poor |
| 2 | | Anhydrous sodium sulfite | | homogeneous | good |
| 3 | | | BN-45 | homogeneous | good |
| 4 | | | BN-42 | homogeneous | good |
| 5 | | Sodium D-isoascorbate | BN-45 | muddy | poor |
| 6 | | Sodium D-isoascorbate | BN-42 | muddy | poor |
| 7 | | Anhydrous sodium sulfite | BN-45 | homogeneous | good |
| 8 | | Anhydrous sodium sulfite | BN-42 | homogeneous | good |

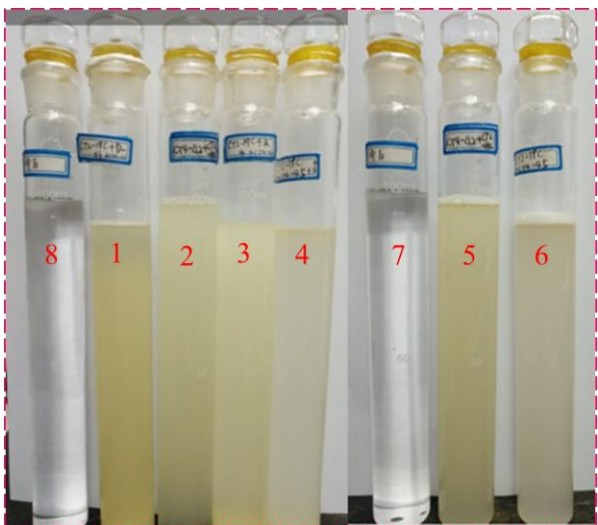

**Figure 8.** Compatibility of additives.

3.2.2. Collaborative Evaluation Experiment

According to the compatibility test results of active additives, group 7 and group 8 are evaluated synergistically in an aqueous solution with a temperature of 60 °C and a $H_2S$: $CO_2$ ratio of 1:2.

Deaeration Rate Experiment

Two groups of deoxidization performance tests are carried out to determine the impact of other additives, including corrosion inhibitors and bactericides, on deoxidization performance. Group one: corrosion inhibitor CT2-19C (5000 ppm) + bactericide BN-42 (2 g/L) + deoxidizer anhydrous sodium sulfite (3 g/L); group two: CT2-19C (5000 ppm) + bactericide BN-45 (2 g/L) + anhydrous sodium sulfite (3 g/L). Figure 9 displays the deoxidation effects in groups 1 and 2. It can be seen that the oxygen content of the two groups of reagents in the solution is controlled below 1 ppm and the deoxidation rate is about 95%, indicating that the deoxygenation effect on the solutions is good in the two groups. In terms of the deoxidization performance, the corrosion inhibitor CT2-19C, bactericides BN-42 and BN-45, and the deoxidizer anhydrous sodium sulfite have good synergy.

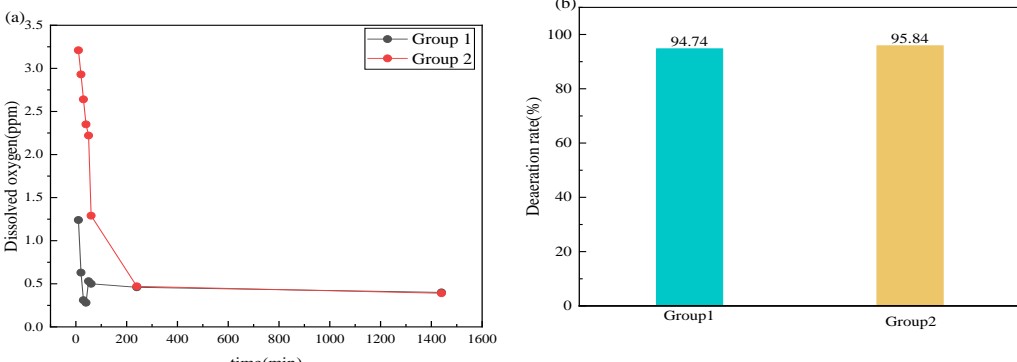

**Figure 9.** Deoxidation effects in groups one and two. (**a**) dissolved oxygen; (**b**) deaeration rate.

Corrosion Inhibition Rate Experiment

According to the compatibility test results, for system one: the corrosion inhibitor CT2-19C (5000 ppm) + bactericide BN-42 (2 g/L) + deoxidizer anhydrous sodium sulfite (3 g/L); System two: CT2-19C (5000 ppm) + bactericide BN-45 (2 g/L) + anhydrous sodium sulfite (3 g/L) are carried out in the corrosion inhibition test.

Figure 10 shows the self-corrosion current density and corrosion inhibition rate of P10ss steel in system one and system two. For group one, it can be seen from Figure 10a that the self-corrosion current is significantly reduced and the corrosion inhibition rate is at 93.41% when the corrosion inhibitor CT2-19C is added, indicating that corrosion inhibitors can greatly slow down the corrosion of P110ss steel. For group two, when the corrosion inhibitor CT2-19C and the deoxidizer anhydrous sodium sulfite were added, the self-corrosion current was greater than that of the corrosion inhibitor CT2-19C alone, implying that the addition of a deoxidizer intensifies the corrosion of P110ss steel. For group three, when the corrosion inhibitor CT2-19C and the fungicide BN-42 were added, the self-corrosion current was greater than that of the corrosion inhibitor CT2-19C alone, indicating that the bactericide weakens the corrosion inhibition of the inhibitor. For group four, compared with the corrosion inhibitor, the corrosion current increases significantly when the corrosion inhibitor CT2-19C, bactericide BN-42, and deoxidizer anhydrous sodium sulfite are added, indicating that the deoxidizer and bactericide weaken the corrosion inhibition performance of the inhibitor.

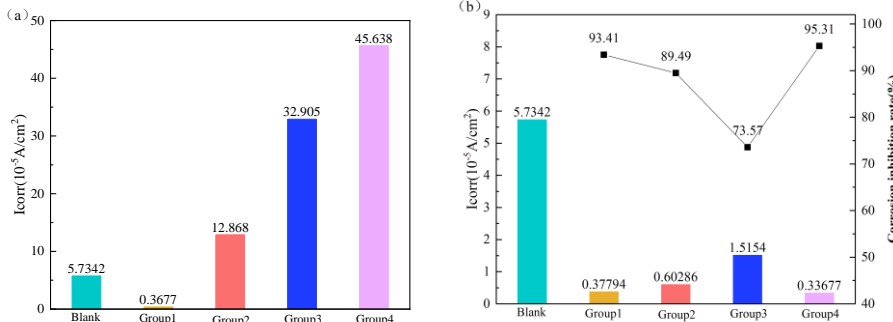

**Figure 10.** Self-corrosion current density and inhibitor efficiency of P110ss steel corroded in system one and 2. (**a**) System one; (**b**) system two.

For group one, it can be seen from Figure 10b that the self-corrosion current decreases significantly and the corrosion inhibition rate is at 93.41% when the corrosion inhibitor CT2-19C is added, implying that the corrosion inhibitor CT2-19C can greatly slow down the corrosion of P110ss steel. For group two, compared with the corrosion inhibitor, the corrosion current increases significantly when the corrosion inhibitor CT2-19C and deoxidizer anhydrous sodium sulfite are added, indicating that the deoxidizer has an

inhibiting effect on the corrosion inhibition. For group three, compared with adding the corrosion inhibitor CT2-19C alone, the corrosion current increases significantly when the corrosion inhibitor CT2-19C and bactericide BN-45 are added, implying that the bactericide has an inhibitory effect on the corrosion inhibition performance of the corrosion inhibitor. For group four, compared with adding the corrosion inhibitor CT2-19C alone, the corrosion current decreases when the corrosion inhibitor CT2-19C, bactericide BN-45, and deoxidizer anhydrous sodium sulfite are added, indicating that the addition of the deoxidizer and bactericide can significantly improve the corrosion inhibition performance of the corrosion inhibitor. According to the experimental results, system two provides better corrosion mitigation than system 1. Therefore, the formulation of the water-based annular protective liquid needs to be designed according to system two in the next step.

### 3.2.3. Composition Design of Annulus Protection Fluid
Ratio Design of Corrosion Inhibitor

P110ss steel is tested under corrosion inhibitors of 5000 ppm, 10,000 ppm, 20,000 ppm, and 30,000 ppm by autoclave, and the suitable concentration of inhibitors is selected. The test design scheme is shown in the following Table 5.

**Table 5.** Experimental scheme for corrosion inhibitor concentration selection.

| Group | Inhibitor Concentration (ppm) | Temperature (°C) | $H_2S$ Partial Pressure (MPa) | $CO_2$ Partial Pressure (MPa) | Total Pressure (MPa) | Time (h) |
|---|---|---|---|---|---|---|
| 1 | 5000 | | | | | |
| 2 | 10,000 | 120 | 1.5 | 3 | 40 | 168 |
| 3 | 20,000 | | | | | |
| 4 | 30,000 | | | | | |

Figure 11 displays the corrosion rate of P110ss steel under different concentrations of corrosion inhibitors. It can be seen that the corrosion rate of P110ss steel in the gas–liquid two-phase system is significantly reduced at the corrosion inhibitor of 30,000 ppm [34]. The corrosion rate in the liquid phase environment can be reduced to 0.0729 mm/y, and the corrosion inhibition rate can reach more than 95%. The corrosion rate in the gas phase environment can reach 0.0723 mm/y, which meets the oilfield corrosion control requirements (lower than 0.076 mm/y), and the corrosion inhibition effect is good. Therefore, 30,000 ppm of the CT2-19C corrosion inhibitor is used for the applicability evaluation.

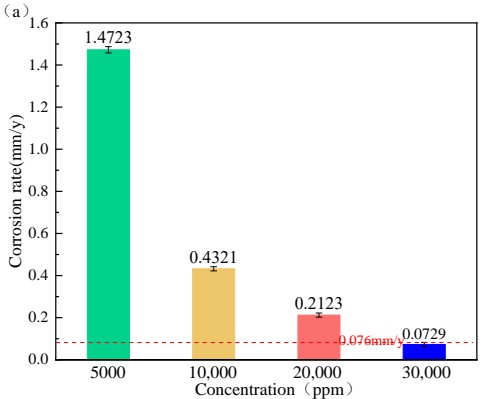
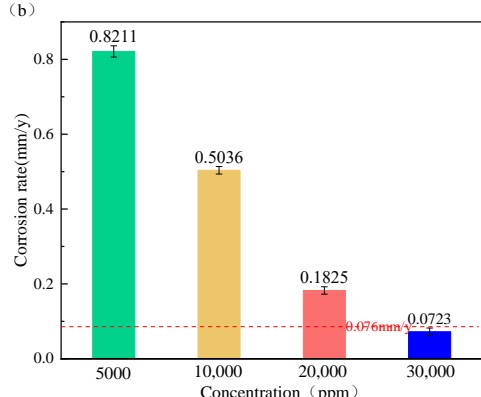

**Figure 11.** P110ss steel corrosion rate at different concentrations of corrosion inhibitor. (**a**) Liquid phase; (**b**) gas phase.

Ratio Design of Deoxidizer

Figure 12 exhibits the measured results of dissolved oxygen in an annular protective solution with different concentrations of the deoxidizer. It can be seen that when the

dosage of oxygen absorber exceeds 3 g/L, the dissolved oxygen content in the annular protective liquid is controlled within 0.3ppm of the oilfield water quality control index (SY/T5889-2010), which has excellent deoxidation performance. With the increase in the oxygen scavenger fill concentration, the oxygen removal rate increases, but when the oxygen removal concentration increases from 3 g/L to 4 g/L, the increase in the oxygen removal rate is not large, the effect is not obvious, and the cost is slightly controlled. The recommended dosage of oxygen absorber is 3 g/L.

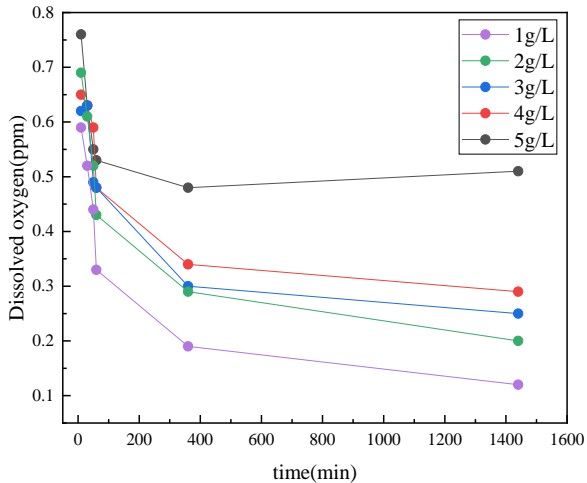

**Figure 12.** Results of dissolved oxygen in annular protective solution under different concentrations of oxygen absorber.

Ratio Design of Bactericide

Table 6 shows the bactericidal performance of BN-45 at different doses in a water-based annular protective solution. The experimental results display that the dosage of fungicide BN-45 is 2 g/L, which can achieve a good bactericidal effect.

**Table 6.** Bactericidal properties in different doses of water-based annular air protection solution.

| Number | BN-45 Dosing (g/L) | Saprophytic Bacteria (CFU/mL) | Iron Bacteria (CFU/mL) | Sulfate Reducing Bacteria (CFU/mL) |
|---|---|---|---|---|
| 1 | 0 | $2.52 \times 10^3$ | $2.52 \times 10^3$ | $0.61 \times 10^4$ |
| 2 | 1 | 2.56 | $5.05 \times 10^2$ | $0.64 \times 10^2$ |
| 3 | 1.5 | 0 | $2.59 \times 10$ | $2.02 \times 10$ |
| 4 | 2 | 0 | 0 | 0 |

### 3.2.4. Physical and Chemical Properties Test

The water-based annular protective liquid including CT2-19C (30,000 ppm) + anhydrous sodium sulfite (3 g/L) + BN-45 (2 g/L) obtained from the above clear water ratio to test the physical and chemical properties of water-based annular protective liquid is used, as shown in Table 7.

**Table 7.** Physical and chemical properties of water-based annular protective fluid.

| Density (g/cm$^3$) | Freezing Point (°C) | pH |
|---|---|---|
| 1.02 | −2.01 | 7.73 |

### 3.2.5. Water-Based Annular Protective Fluid Formulated by Formation Water

It can be known that clear water-based annular protective liquid includes CT2-19C (30,000 ppm), anhydrous sodium sulfite (3 g/L) and BN-45 (2 g/L) from the above, it is

necessary to further design the formula of water-based annular protective fluid prepared by field formation water (Table 2).

Table 8 and Figure 13 show the compatibility test scheme and results between various additives and formation water diluted with different proportions of clear water. clear water It can be seen that Group nine and Group eleven have good compatibility, and 50% formation water + 50% clear water + 30,000 ppmCT2-19C + 3 g/LD sodium isoascorbate + 2 g/LBN-45 is recommended. If the site conditions cannot be diluted, it is necessary to further screen organic deoxidizers that are easy to dissolve, compatible and environmentally friendly.

**Table 8.** Compatibility test scheme and results between various additives and formation water diluted with different proportions of clear waterclear water.

| Group | Formation Water | Water | Corrosion Inhibitor | Deoxidizer | Bactericide | Phenomenon (30 min,60 °C) | Evaluation Results |
|---|---|---|---|---|---|---|---|
| 1 | 100% | | CT2-19C | Anhydrous sodium sulfite | | precipitate | Poor |
| 2 | | | | Sodium D-isoascorbate | | precipitate | Poor |
| 3 | | | | Anhydrous sodium sulfite | BN-45 | precipitate | Poor |
| | | | | Sodium D-isoascorbate | BN-45 | precipitate | Poor |
| 4 | 75% | 25% | | Anhydrous sodium sulfite | | precipitate | Poor |
| 5 | | | | Sodium D-isoascorbate | | precipitate | Poor |
| 6 | | | | Anhydrous sodium sulfite | BN-45 | precipitate | Poor |
| 7 | | | | Sodium D-isoascorbate | BN-45 | precipitate | Poor |
| 8 | 50% | 50% | | Anhydrous sodium sulfite | | precipitate | poor |
| 9 | | | | Sodium D-isoascorbate | | homogeneous | Good |
| 10 | | | | Anhydrous sodium sulfite | BN-45 | precipitate | Poor |
| 11 | | | | Sodium D-isoascorbate | BN-45 | homogeneous | Good |

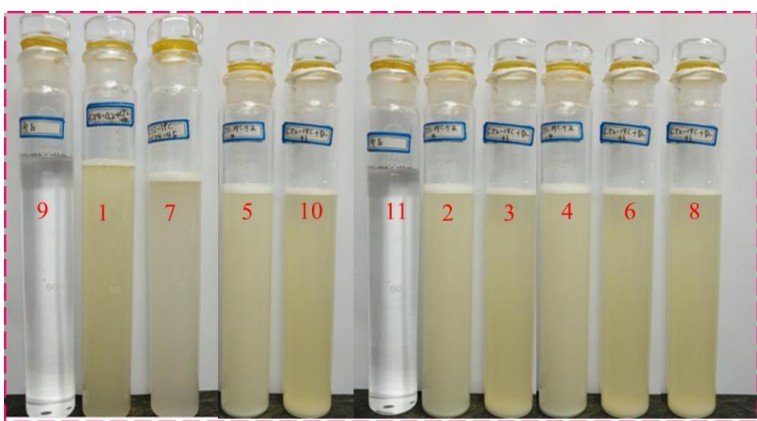

**Figure 13.** Compatibility test between various additives and formation water diluted with different proportions of clear water.

3.2.6. Corrosion Experiment of Water-Based Annular Protective Liquid

According to the ratio design results of the above water-based annular protective fluid, the formula of the formation water-based annular protective fluid is system one: clear water +

corrosion inhibitor CT2-19C (30,000 ppm) + deoxidizer anhydrous sodium sulfite (3 g/L) + bactericide BN-45 (2 g/L); system two: 50% formation water + 50% clear water + corrosion inhibitor CT2-19C (30,000 ppm) + deoxidizer D-sodium isoascorbate (3 g/L) + bactericide BN-45 (2 g/L).

The effect of water-based annular protective liquid is evaluated by using an HTHP autoclave to carry out weight-loss tests, and the corrosion evaluation results of P110ss in a gas–liquid two-phase environment are obtained. Test conditions: total pressure of 40 MPa, temperature of 160 °C, $H_2S$ partial pressure of 1.5 MPa, $CO_2$ partial pressure of 3 MPa, test time of 72 h, in a corrosion solution. As can be seen in Figure 14, both system one and system two have good corrosion inhibition effects ($\leq$0.076 mm/y) in the gas and liquid phases, and the effect of system one is better than that of system two.

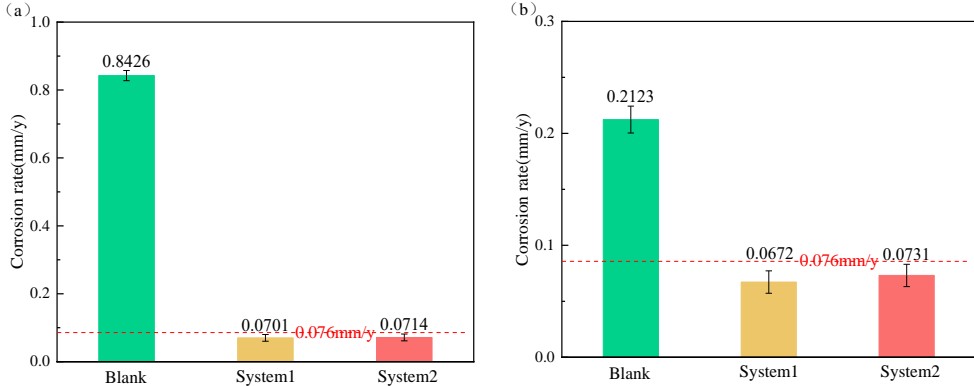

**Figure 14.** Corrosion rate of P110ss steel immersed in solution added with water-based annular protective liquid. (**a**) Liquid phase; (**b**) gas phase.

Figure 15 displays the macro- and micro-morphology of P110ss in the water-based annular protective solution. The water-based annular protective solution design includes system one: clear water + corrosion inhibitor CT2-19C (30,000 ppm) + deoxidizer anhydrous sodium sulfite (3 g/L) + bactericide BN-45 (2 g/L); system two: 50% formation water + 50% clear water + corrosion inhibitor CT2-19C (30,000 ppm) + deoxidizer D-sodium isoascorbate (3 g/L) + bactericide LBN-45 (2 g/L). It can be seen that the water-based annulus protective liquid is uniformly and continuously covered on the surface of the sample in both the gas and liquid phases after the addition of system 1 [34]. According to the results obtained, the surface of the P110ss steel is smooth, and no obvious traces of corrosion are observed (Figure 15a–d). When system two was added to the solution, the surface of the P110ss steel sample in the gas–liquid two-phase was relatively flat, and slight local corrosion traces were observed. According to the EDS results (Figure 16), the corrosion products contain a large amount of Ca. It is inferred that the corrosive pitting is caused by the scaling of corrosion media, such as $CO_2$ and $H_2S$, with the formation water containing high Ca ions. Because formation water contains calcium ions, the long-term use of water-based annulus protection fluid prepared from formation water may lead to under-deposit corrosion. In order to reduce the risk of under-deposit corrosion, it is recommended to use water-based annulus protection fluid prepared with clear water.

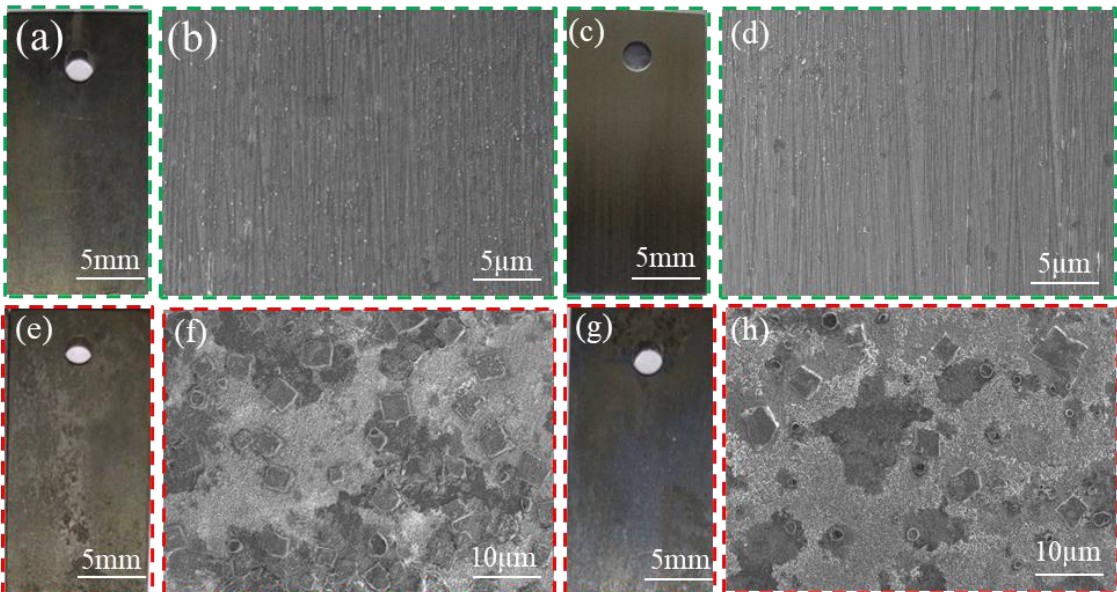

**Figure 15.** Macroscopic morphology and micromorphology of P110ss in corrosive solution prepared with water-based annular protective solution prepared with clear water (**a–d**) and formation water preparation (**e–f**) (liquid phase: (**a,b,e,f**); gas phase: (**c,d,g,h**)).

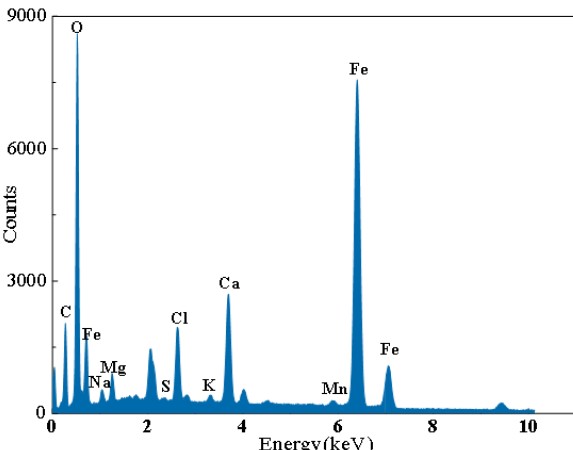

**Figure 16.** EDS results of corrosion products of P110ss steel in formation water-based annular protective fluid.

## 4. Conclusions

In this paper, the single-agent screening, compatibility, and synergy evaluation of active additives in a water-based annular protective fluid were successively carried out through electrochemical testing, the ratio of a water-based annular protective liquid was optimized, two sets of water-based annular protective fluid formulas were formed, and then the protective effect of the new water-based annular protective liquid was evaluated by simulating the field working conditions of a high-temperature autoclave, so the following conclusions were obtained:

(1) The formula of one clear water-based annular protective liquid includes clear water, a corrosion inhibitor CT2-19C (30,000 ppm), bactericide BN-45 (2 g/L), and deoxidizer anhydrous sodium sulfite (3 g/L), and the other contains 50% formation water, 50% clear water, retarder CT2-19C (30,000 ppm), bactericide BN-45 (1 g/L), and deoxidizer D-sodium isoascorbate (3 g/L).

(2) Annular protective fluid prepared by formation water is easy to scale because formation water contains $Ca^{2+}$ and $Mg^{2+}$ ions, and the risk of long-term use is greater. The produced

wells are recommended to use water-based annular protective liquid prepared with clear water, including CT2-19C corrosion inhibitor (30,000 ppm) + anhydrous sodium sulfite (3 g/L) + BN-45 bactericide (2 g/L).

(3) The density of water-based annular protective liquid prepared with clear water is 1.02 g/cm$^3$, and its freezing point is $-2.01$ °C. The water-based annular protective liquid controls the dissolved oxygen content in the injected water within 0.3 ppm.

(4) The corrosion inhibition rate of clear, water-based annulus protection fluid exceeds 90%, and the corrosion rate of steel in the gas phase is lower than the oilfield corrosion control index (0.076 mm/y). The water-based annular protective fluid meets the anti-corrosion requirements of actual production and working conditions, so it can be popularized and applied in sour gas field production.

**Author Contributions:** Conceptualization, X.H. and D.Z.; methodology, Q.L.; software, J.C.; validation, N.J., R.Z. and W.C.; formal analysis, Q.L.; investigation, J.C.; resources, D.Z.; data curation, Q.L.; writing—original draft preparation, X.H.; writing—review and editing, D.Z.; visualization, X.H.; supervision, D.Z.; project administration, L.D.; funding acquisition, D.Z. All authors have read and agreed to the published version of the manuscript.

**Funding:** The authors acknowledge the support from the Sichuan Science and Technology Program (21JCQN0066) and the support from the National Natural Science Foundation of China (No. 51774249).

**Data Availability Statement:** Not applicable.

**Conflicts of Interest:** The authors declare no conflict of interest.

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
