# Peer review of "Design of Multifunctional and Efficient Water-Based Annulus Protection Fluid for HTHP Sour Gas Wells"

_processes, doi:10.3390/pr11010171_

Round 1

Reviewer 1 Report

The authors presented the process of selecting a water-based product to reduce the rate of corrosion in wells where, apart from crude oil, there is an acidic environment caused by the presence of CO2 and H2S. The product was optimized in terms of the selection of corrosion inhibitor, biocide and deoxidizer and their concentration.

The manuscript has no scientific but practical significance. It does not contain new ideas. However, it may be helpful for readers who deal with corrosion protection in similar conditions.

The whole narrative of the authors is quite simplistic and chaotic. Add to this the errors in the English language, reading the manuscript makes it somewhat difficult. I would like the authors to work on the clarity of the entire manuscript.

Some test results, e.g. electrochemical, are missing, only the final results in the form of corrosion rate are presented. There is no information about the equipment used for electrochemical tests. Impedance studies were mentioned in the experimental methodology section (page 4) but no results were mentioned afterwards. So this information is unnecessary. It is not described how the corrosion rate was determined from the polarization curves. Without these details, the use of the procedures presented is problematic for readers. Used inhibitors, biocides, and standards refer to the Chinese market, short information in this area for other readers would be useful.

There are a lot of syntax and style errors, punctuation and other errors. This requires careful correction, e.g.

a) no spaces (l. 70), "including 1.5MPaH2S and 3MPaCO2"

b) l. 72-73, "Therefore, the annulus pressure is 40MPa, the temperature of 160℃, and the corrosion environment is harsh"

c) repetition of the word, (l. 101), "standard, traditional three traditional three electrode"

d) (l. 108) 100000Hz is 100kHz

e) (l. 138) "Wheres" unless where:

f) (l. 238-239) "has a inhibiting effect on the corrosion inhibition

effect of the inhibitor" ?

g) (l. 260), "of 1200ppm", there is no such value in Figure 10

h) table 6, Chinese letters

i) table 8, some words not visible

j) (l. 338) the word "gas" is missing

k) the authors sometimes use "Fig." and other times "Figure" - this requires unification

l) the authors use the phrase "sustained release rate" (e.g. Figures 6 and 9), in my opinion, it is "inhibitor efficiency"

Reviewer 2 Report

I suggest to double check the language and add a section of results and discussion before the conclusion section. Kindly follow the highlights and act accordingly.

Reviewer 3 Report

The research content of the paper is meaningful, but there are some needs to be improved, as follows:

1. The language of the paper needs to be carefully revised. For example, the use of the word "screening" is unreasonable.

2. In the Introduction, more discussion need to be described. The significance of the study also needs to be specified.

3. The structure of the paper is suggested to be improved. It is unnecessary for the manuscript to be divided into so many parts. The experimental part is suggested to be optimized, and the current version is chaotic. The experimental results were divided into additive research and protection fluid evaluation.

4. The test results in Table 3 and Table 4 are not clear, how to judge whether the compatibility is qualified? Are any pictures or data added?

5. The explanation in Figure 11 is too simple. From the author's description, the reason for choosing 3g/L is inaccurate.

6. The units in Table 6 are in Chinese, which needs to be modified.

7. The display in Table 8 is incomplete, which needs to be modified. Moreover, the test results of compatibility are not clear. Are any pictures or data added?

8. The conclusion needs to be improved.

Round 2

Reviewer 1 Report

The authors corrected a large part of the detailed changes I suggested, although in general the whole article still bears the hallmarks of chaoticity, rather little has been improved in this respect. At the same time, it requires careful correction of the English language both in terms of style and punctuation, this also applies to newly introduced fragments of text. Further information in the experimental part is quite general.

There are some strange fragments, e.g. lines 124-125, which means the insertion: "disturbance of 5mV applied after the working electrode reaches a stable open circuit potential in the test solution" (probably this is a remnant of the previous version?) or lines 128-129: "The temperature of the electrochemical experiments is 60℃." Does the experiment have a specific temperature? Rather: Tests were conducted at 60℃ or something similar. The entire text needs to be reviewed in this respect. The authors corrected the captions of the old figures 6 and 9 (in the current version 7 and 10) using the term "inhibitor efficiency" but the previous incorrect term "sustained release rate" remained in the axis descriptions in the figures. In the caption of Fig. 7 there is no information that the corrosion current density is also presented, The unit of corrosion rate is "mm/y" in some places, "mm/a" in others, it needs to be standardized, it should be "mm/y" everywhere.

These types of manuscript corrections are necessary for the article to be correctly understood by readers.

Author Response

Thank you for your helpful comments.

According to the reviewer, we have modified the content of lines 124-130. The issues of temperature and expression confusion mentioned in the comments have been revised one by one and highlighted in the revision. The expression of the corrosion inhibition rate in the paper has been checked and revised in the whole paper, as shown in Figure 7 and Figure 10.

Reviewer 2 Report

You paper looks better.

Author Response

Thank you for your comments and recognition.

Reviewer 3 Report

The manuscript can be published in the current version.

Author Response

(The authors gave the same response as above.)
